# Exploring the Potential of AI-Assisted Technology in Joint Range-of-Motion Measurements: A Reliability Study

**DOI:** 10.3390/medicina61010119

**Published:** 2025-01-14

**Authors:** Gisoo Lee, Eric W. Tan

**Affiliations:** 1Department of Orthopedic Surgery, Chungnam National University School of Medicine, Daejeon 34134, Republic of Korea; 2Department of Orthopedic Surgery, The Keck School of Medicine of USC, Los Angeles, CA 90033, USA; eric.tan@med.usc.edu

**Keywords:** POM-checker, goniometry, reliability and validity, artificial intelligence, standing position

## Abstract

*Background and Objectives*: Measuring joint range of motion (ROM) is essential for diagnosing and treating musculoskeletal diseases. However, most clinical measurements are conducted using conventional devices, and their reliability may significantly depend on the tester. This study implemented an RGB-D (red/green/blue-depth) sensor-based artificial intelligence (AI) device to measure joint ROM and compared its reliability with that of a universal goniometer (UG). *Materials and Methods*: A single-center study was conducted from January 2022 to December 2022 on participants visiting the Chung-nam National University Hospital to compare the reliability of the RGB-D sensor-based AI device with that of the UG for measuring ROM. The ROM of the shoulder, hip, and lumbar spine joints was measured in 35 healthy participants in our hospital. The ROM was measured during active motion by the participants in the standing position. The ROM was measured twice consecutively using the RGB-D sensor-based AI device, and the mean values were obtained along with other values. A clinician also measured the ROM twice using a UG. Bland–Altman analysis was performed to evaluate the reliability of the measurements, which was assessed using intra-class correlation coefficient (ICC). An ICC value greater than 0.90 indicates excellent reliability. *Results*: Both methods achieved good-to-excellent intra-test reliability results (ICC > 0.75) for all the joints, with the reliability being slightly higher for the RGB-D sensor-based AI method than for the UG measurements. Moreover, for both methods, the inter-test reliability was higher than good (ICC > 0.75) for shoulder and lumbar joint ROM measurements but lower than good (ICC < 0.75) for hip ROM measurements. *Conclusions*: This study compared the efficacies of the RGB-D sensor-based AI method and UG in measuring ROM. In the future, this RGB-D sensor-based AI method should be technologically improved, and the measurement methods and protocols should be standardized.

## 1. Introduction

A crucial index in clinical practice related to the musculoskeletal system is the range of motion (ROM) of joints [1]. The evaluation of the range and patterns of joint movement is considered important by many clinicians to diagnose and treat patients with musculoskeletal problems. Currently, clinicians use various measurement devices, such as goniometers, inclinometers, tape measures, and marker attachment systems, to make these assessments. Each method has certain advantages and disadvantages, and the measurement reliability of a method is a crucial factor in selecting it for use. Clinicians must accurately know how any change in ROM over a period corresponds to actual changes in the patient. For example, in clinical measurements, even a small difference in angle, such as less than 1°, could change the evaluation criteria [2,3]. Currently, the universal goniometer (UG) is the standard tool used by clinicians to measure ROM [4]. The reliability of UG measurements is simply the consistency or repeatability of the ROM measurements, that is, whether the application of the instrument and procedures consistently produce the same measurements under the same conditions. The UG is reported to possess excellent inter- and intra-test reliability for assessing the ROM of the upper extremities [5]. In contrast, the results are unsatisfactory or insufficient when other joints are considered. Moreover, bias in the measuring instrument or process can occur because of several factors. The reliability of the inter- and intra-testers in ROM measurement could be unsatisfactory in specific movements of some joints even with a standardized method [1,6,7,8]. The UG requires two hands to manipulate the instrument, which must be positioned accurately for alignment and requires clear visual estimation of a measurement reading. It can suffer from high variability owing to manual operation [9]. Furthermore, the subjectivity of the tester may influence the measurement process. Although the measurement of active ROM is important for assessing the functional status of a joint, most ROM measurements are performed for passive movement while the participant is lying down because active ROM measurement still requires an accurate standard protocol for measurement; otherwise, it presents higher errors. In addition, the information provided by UG measurements is limited because the results are expressed only as an angle value without any specific details about the functional state, power, or speed. These limitations have motivated the development of novel measurement methods. Accordingly, the reliability and validity of these new measurement methods should be evaluated [10,11].

Based on the above considerations regarding the measuring process using the UG, we assessed the ROM of joints using an artificial intelligence (AI) algorithm based on a three-dimensional (3D) RGB-D (red/green/blue-depth) sensor [12]. AI was integrated to compensate for the limitations of RGB-D devices [13,14]. The RGB-D sensor enables precise and objective measurement of the movements of body joints. In our process, we used the supplementary AI algorithm to recognize and supplement the phenomenon of body obscuration in some movements, which is the main shortcoming of the RGB-D measurement method, or errors in the measurement and analysis process. Additionally, the algorithm ensured the correct starting angle during measurement and compensated for the maximal angle. In this study, the AI-based and UG methods were simultaneously implemented to measure the ROM of the participants’ shoulder joints. The measurements of this joint’s ROM have been validated by several studies [9,15]. Moreover, the same methods have also been implemented for measuring the ROM of the hip and lumbar spine joints [12,14]. However, there have been few clinical research attempts similar to the above method for the hip joint and lumbar spine to date, and no satisfactory results have been reported. Therefore, this study evaluated the reliability of active ROM measurements for the shoulder, hip, and lumbar spine joints by combining image acquisition and an AI algorithm using an RGB-D sensor. The primary objective was to analyze the reliability of the two measurement methods. The secondary objective was to determine the advantages and disadvantages of the 3D RGB-D sensor-based AI method for measurements in the standing position and the improvements required for clinical application. The results can provide insights into determining the advancements achieved in current clinical musculoskeletal field applications and provide directions for improvement. The hypothesis of this study was as follows: the developed AI measurement device based on a 3D RGB-D sensor provides measurements that are as reliable as those made by clinicians using a UG.

## 2. Materials and Methods

This study was conducted on healthy volunteers who visited the Chung-nam National University Hospital and agreed to participate in the study after being provided adequate information about it. The participants were enrolled between January 2022 and December 2022. They were selected from a convenience sample. This study followed the guidelines for reporting reliability and agreement studies [16]. All participants provided informed consent for the trial. The manual included in the consent form provided easy-to-understand texts, videos, and pictures for all measurement movements. The study was approved by our institute’s review board (approval number: CNUSH-2020-10-017).

### 2.1. Participants

The participants involved in this study underwent screening and physical examination by an orthopedic surgeon (G.-S.L). They were individuals with normal shoulder, hip, and spinal joints. Individuals who had undergone previous surgery or had a disease were excluded. Furthermore, as the study was conducted in the standing position, participants with joint function problems, such as discomfort in standing upright or weakness in one leg, were excluded from this study. Weight and height were measured with participants in light clothing. Body mass index (BMI) was calculated as weight in kilograms divided by the square of the height in meters. Thirty-five healthy adults were included in the study after applying the inclusion and exclusion criteria. The ROMs of the shoulder and hip joints of each participant were measured on both joints, and the ROM of the lumbar spine was measured once. Thus, the shoulder and hip ROMs were measured in 70 cases, and the lumbar spine ROMs were measured in 35 cases. For the intra- and inter-test reliabilities, 16 ROMs of the three joints were measured four times each.

The participants were assigned identification numbers, and their eligibility was evaluated based on specific enrollment criteria. The inclusion criteria were as follows: the participants must be in the age group of 20–59 years, and they must willingly agree to participate in the study and provide their written informed consent after being given a clear explanation of the objectives and characteristics of the clinical study. Furthermore, eligible participants should have no preexisting musculoskeletal disorders of the shoulder, hip, or spinal joints and should be able to stand for approximately 10 min.

The exclusion criteria were as follows: (1) pregnancy; (2) individuals with mental disorders such as nervousness, claustrophobia, or panic disorders; (3) individuals with cardiovascular disturbances including orthostatic hypotension or nervous system problems, such as dizziness; (4) those with existing musculoskeletal disorders or mechanical abnormalities, such as fractures or acute sprains in the joints; (5) individuals experiencing severe disability or significant pain around those joints; (6) individuals with prior surgery or severe injuries to the measured joint; (7) participants with a BMI > 35 and World Health Organization’s criteria for class 2 obesity were excluded to minimize the possibility that joint ROM was limited because of excess adipose tissue [17]; and (8) individuals who insisted on wearing loose clothes and were afraid of being captured by the AI algorithm based on a 3D RGB-D sensor or fearful of slight body touch that could occur during UG measurement.

### 2.2. ROM Measurement Protocol in Both Methods

Four repetitions of 16 ROMs were performed for three joint assessment tests for each participant. The participants were instructed to perform all movements of the shoulder, hip, and lumbar spinal joints in a standing position. All movements were explained using videos and pictures for approximately 15 min, and the participants were asked to stretch their shoulders, hips, and spinal joints for approximately 3 min before measurements. All participants were positioned in front of a white and clean wall, and their joint ROMs were measured in an environment with no obstacles and a flat floor. The participants stood naturally, with their feet shoulder-width apart and their toes pointed straight ahead, facing forward (Figure 1). The tester verified that the shoulders and pelvis were parallel to the ground. If their posture was unstable or incorrect, the participants were asked to correct it. After completing one set of measurements, the participants were allowed to sit down or move around for a while if they wanted to. Subsequently, the above procedures were verified, and the test was continued.

An orthopedic surgeon with 12 years of experience performed the measurements using the UG. The first measurement was performed using the UG. The second and third measurements were performed using the RGD-D sensor AI device. A physician assistant with 5 years of experience performed the measurements. The third measurement was performed 30 s after the second measurement using the AI-based device. The fourth measurement was performed using the UG. The procedure was performed twice at the same location. The overlap or influence of the first UG measurement on the second measurement was avoided by implementing the RBG-D sensor AI device method between the UG measurements. The results of the first and second UG measurements were recorded on paper by an orthopedic surgeon, and the first result was stored elsewhere immediately after the measurement to avoid any biases in the second measurement. Between the first and fourth measurements, the UG tester was asked to move to another location and was not allowed to observe the measurements obtained by the RGB-D sensor AI device. After completing the measurement using the AI device, the physician assistant moved to the next room, and the orthopedic surgeon was notified to perform the 4th measurement within 1 min after the 3rd measurement.

The participants were instructed to move all joints to the maximum possible range. The testers provided verbal instructions regarding the initial and terminal actions to be performed for each movement and communicated the desired actions to the participants for active ROM. The participants were repeatedly reminded to keep their bodies in an upright position and as stable and balanced as possible when moving their joints and to keep performing these movements as similar as possible in all four iterations. Each joint was actively moved to its full extent, and the endpoint measurements as the maximum values were obtained using both methods. If the participants lost their balance or tilted at the maximum ROM, they were requested to perform the test again. Each movement started at the zero position and returned to the zero position after the movement was performed. The subsequent movements began after the tester established and confirmed the zero position. The tester continuously monitored the movements. The entire measurement process was monitored by a 3rd person, a researcher (M-.S.J.), who ensured that the regulations were followed.

Regarding the joint measurement method, the tester instructed the participants as in the previous measurements. The RGB-D sensor AI device was used as described below. For the measurements using the UG, the tester instructed the participants to return to the starting position after the measurement was complete. During the measurements using the UG, the angle measurement could be completed relatively quickly because the tester waited in front of the participant. The following instructions were provided to the participants by the testers:Shoulder
−Stand upright for the (right/left) shoulder.−Forward flexion (FF) and extension: Raise your arm straight (forward/backward) as much as possible and bring your arm toward the midline of the body.−Abduction and adduction: Raise your arm (outward/inward) as much as possible (bring your arm up sideways) and bring your arm toward the midline of the body.−Abduction90 external rotation (ER) and internal rotation (IR): Raise your arm to the side, keep it horizontal, and keep your elbow at 90° with your hand facing forward. Then, turn your forearm (upward/downward) as much as possible, and bring your arm toward the midline of the body.−FF90 ER and IR: Raise your forearm forward, keep it horizontal, and keep your elbow at 90°. Then, rotate your hand (outward/inward), and bring your arm toward the midline of the body.Hip
−Stand upright and place both hands on your waist. For the (right/left) hip−Flexion and extension: With your knees straight, raise your leg (forward/backward) as much as possible and lower your leg.−Abduction and adduction: Raise your leg (outward/inward), swing your leg away from the midline as much as possible, and lower your leg.Lumbar spine
−Stand upright with extended knees.−Flexion and extension: Bend your upper body (forward/backward) as much as possible and stand upright again.−Left and right lateral flexion: Bend your upper body (to the left side/right side) as much as possible and stand upright again.

### 2.3. UG Measurement

An XTender™ HiRes^®^ 360° ISOM goniometer (SKU: BA12-1034HR Brand: Baseline MPN/Model:12-1034HR, United States; UOM: Each UPC: 714905076724), which can extend from 12.5″ to 33″ was used for the measurements (Figure 1). For the UG measurements, all joint movements started at 0°, and the joints were moved to their maximum ROM, which was measured. When using the UG, the stationary arm was aligned parallel to the base of the standardized angle for the starting point of the measurement. The UG axis of rotation was aligned over the center of the hinge joint, and the moving arm was aligned parallel to the angle formed by the arm to identify the maximum extension. By aligning the stationary and movable arms of the device with specific bony landmarks on either side of the joint, the full extent of joint mobility can be measured in degrees. The most commonly used reference values for joint ROM are those published by the American Academy of Orthopedic Surgeons [4] or the Kendall values [18]. Furthermore, for each joint, surface anatomy, and bony landmark, the methods introduced by Clarkson et al. [19] and Reese et al. [20] were applied to the standing position. After each measurement, the UG was reset to 0°. The UG scale (0–360°) provided readings in increments of 1°. Thus, the measured angle was approximated to the nearest degree.

### 2.4. Three-Dimensional RGBD Sensor and AI Analysis (POM-Checker Measurements)

The 3D RGB-D sensor AI device used in this study was the POM-Checker (Team Elysium Inc., Seoul, Republic of Korea; POM is a portmanteau of posture and ROM). The POM-Checker has obtained a product license (license number: 18-4334) for conformity assessment under the classification code A30130.01(2), specifically for isokinetic testing and evaluation systems. This license was granted by Korea Testing Certification, which has been authorized as the status of the National Official Professional Testing Research Institute in South Korea.

A physician assistant operated the POM-Checker software (version 1.5.1) process and guided the participants’ movements. The participants stood in front of the POM-Checker, and behind them was a white, clean wall in a relatively large and bright room illuminated by a light-emitting diode-led fluorescent light. All measurements were obtained in the standing position with the participants looking directly at the POM-Checker and facing forward. The participants stood at a distance of 2 m from the POM-Checker, and tape markers were placed on the floor to ensure that the same distance was maintained for all participants. The POM-Checker was placed on the mark with tape on the floor, as it is a movable device, and measurements were performed in the same place. The angle was measured using the POM-Checker to one decimal place. Some participants with high flexibility had ROM angles that were beyond the normal range or the expected values. Therefore, we predefined a maximum angle value for some of the shoulder ROM measurements. For example, the maximum values were predefined at 180° for FF and abduction and 90° for FF90-ER and Abduction90-ER. For the other joints, the maximum value was not limited. This was also applied when the ROM was measured using the UG. For the shoulder ROM, the values measured using a UG above the normal limit were changed to the normal maximum value. The POM-Checker was adjusted accordingly. This information was not given to the participants before the measurement.

The POM-Checker captures RGB-D images based on the time-of-flight method, which measures the time for infrared rays to reach an object and then reflect and return. This method can be affected by external light sources or the test participant’s clothing. Due to challenges in obtaining high-quality images in some environments, this study was conducted under laboratory conditions, as described previously. For example, if any participant had worn very large or baggy clothes, they were asked to change their clothes. By combining existing body recognition technology based on depth images using 3D RGB-D sensors with AI technology to analyze and correct RGB images received from 3D sensors, we expected the body ROM measurement device to be stable, irrespective of the surrounding environment, and independent of bias related to the testers [15].

The POM-Checker comprises a main computer, monitor, and Kinect v2. Kinect v2 is robust to artificial illumination and sunlight [21] and can accurately track the human body [22]. Studies have found Kinect v2 to be valid and reliable for measuring ROM [23,24]. The POM-Checker utilizes 3D cameras, and the AI system can identify joint positions, enabling the real-time tracking of joint movements and relative positions in 3D space [25]. RGB images and depth information are acquired from the RGB-D sensors. The software uses a 3D sensor (RGB-D sensor) to identify the 3D locations of the body parts, enabling automatic joint angle measurements. It can track 25 body joints [26]. The POM-Checker recognizes the participants from the depth image input of the 3D sensor and automatically recognizes their main joint position, quantitatively measuring the joint ROM. An open-access convolutional neural network, particularly a deep learning algorithm [27,28], was implemented to recognize the shoulder, hip, and lumbar spine from the captured RGB-D images. The deep learning model detected human figures in the RGB images and estimated the positions of the joints in two dimensions. Based on these data, depth images were used to obtain the corresponding joint positions on the 3D surface of the human body. These 3D surface joint positions were processed using another deep learning model to accurately estimate the actual 3D locations [29].

The angles between the joint connections and reference vectors were also measured. The ROM was determined by calculating a measurement vector by connecting the joints of the body to be measured and measuring the angle at which the vector was projected onto the frontal, sagittal, or horizontal plane (Figure 2). The measurement vectors for this study were the vectors connecting the center of the shoulder and elbow for the shoulder ROM, the hip and knee for the hip ROM, and the midline of the pelvis (spine base) and middle spine (spine mid) for the lumbar spine ROM [26]. To reduce measurement noise and increase accuracy, we applied the exponential moving average method [30], which is a simple and widely used filtering method. In a previous study [15], the POM-Checker had demonstrated higher reliability than the 3D motion capture analysis system for the shoulder area.

### 2.5. Statistical Analysis

Thirty-five individuals participated in the study. Patient characteristics were also analyzed. Data from 70 cases involving the shoulder and hip and 35 cases involving the spine were obtained using the UG and POM-Checker. Each measurement method was performed twice. The average and standard deviation (SD) of the two measured values were analyzed. Paired *t*-tests were performed to evaluate the systematic differences between the UG and POM-Checker measurements.

Bland–Altman plots and the identified mean difference, SD of the mean difference, standard error of the mean difference, and 95% confidence interval (CI) for the limits of agreement between the UG and POM-Checker were calculated. First, the average and SD corresponding to the difference between the two measured values were calculated (bias, SD-bias). Second, Bland–Altman analysis and intra-class correlation coefficient (ICC) (2,1) analysis were performed to determine whether there was a significant difference between the results obtained from using the UG and POM-checker twice (intra-test reliability). Bland–Altman analysis and ICC (2,2) analysis were conducted to determine whether the average values of the UG and POM-Checker measurements had a significant difference (inter-test reliability) [31].

As a reference, the Bland–Altman plots [32] represent the results when measurements are repeatedly obtained from the same groups of participants, and the true value is constant for each of the measurements. Agreement is the ratio of values that fall outside the 95% CI of the difference when the average of the two values and the difference between the two values are displayed as a scatterplot in a Bland–Altman plot. The higher the ratio, the higher the degree of agreement between the two values. The ICC [33] assesses how similar measurements obtained from the same group are. This index was used to estimate the degree of agreement between the values measured using the UG and POM-Checker. Concurrent validity and inter-test reliability were established using the two-way mixed ICC for absolute agreement [34,35]. ICC with a 95% CI was used to show the intra-test reliability of the measurements.

Inter- and intra-test reliability was considered poor for ICC values less than 0.50, fair for values between 0.50 and 0.75, good for values between 0.75 and 0.9, and excellent for values greater than 0.90. The ICC for reliability should be greater than 0.90 to ensure reasonable validity [35,36].

The measured data were analyzed using R software, v.4.1.0 (R Foundation for Statistical Computing, Vienna, Austria), and were analyzed at a significance level of 0.05.

## 3. Results

The demographic data of the participants is summarized in Table 1. Among the enrolled participants (*n* = 35), the number of females (*n* = 20, 57.1%) was slightly higher than the number of males (*n* = 15, 46.9%). All the participants were Asian (Korean). The average age of the participants was 38.68 (SD: 7.89, 24.6–53.6) years, with the female participants being marginally older than the male participants (*p* = 0.07). The average BMI of the participants was 23.39 kg/m^2^ (SD: 2.41, 19.4–28.1), and it was significantly higher in the males than in the females. The mean height was 168.21 cm (SD: 7.14, 152.4–185.2). Thirty-two participants were right-dominant (91.4%).

ROM measurements of three joints (shoulder: eight movements, hip: four movements, and lumbar spine: four movements) were taken for all 35 participants. The data of the shoulder and hip joint ROMs of the left and right sides of the participants comprised 70 cases, and the lumbar spine ROM measurements comprised 35 cases. Each participant performed four sets (UG: first and fourth sets and POM-Checker: second and third sets) of repetitive movements. The average time taken per participant for the measurements was 25.8 [20,21,22,23,24,25,26,27,28,29,30,31,32,33,34] min. The time gap between the UG measurements (the end of the first and the beginning of the fourth sets) was 13.5 [10,11,12,13,14,15,16,17,18,19,20,21] min on average.

A comparison was performed between the two values measured using each method, and the mean values and SD were calculated and analyzed for the two methods. For most joints, no statistically significant difference was observed between the average values measured twice using each device. However, some differences were observed for the hip joint. The mean values of the hip ROM obtained using the POM-Checker were significantly higher than those obtained using the UG for the abduction and hip extension cases (*p* = 0.044 and 0.000); however, the values obtained for the adduction case using the UG were significantly higher than those obtained using the POM-Checker (*p* = 0.000). Furthermore, although the hip joint usually has a relatively smaller ROM than that of other joints, the SD of the hip joint ROM was relatively higher than the SDs of the ROMs of other joints (Table 2 and Table 3).

As mentioned previously, we focused predominantly on the ICC values to evaluate measurement reliability. The intra-test ICC (2,1) scores for all variables are summarized in descending order in Table 2 and Table 3, and the inter-tester ICC (2,2) scores are summarized in Table 4. Table 2, Table 3 and Table 4 also include the 95% confidence intervals, standard error of the mean (SEM), and minimal detectable change (MDC95). Figure 3, Figure 4 and Figure 5 show Bland–Altman plots to visualize the agreement or consistency between the two measurement methods.

Regarding intra-test reliability, the ICC (2,1) for values measured twice by each tester indicates relatively high reproducibility (reliability and agreement) for both the UG and the POM-Checker measurement methods, because a majority of the ICC values was “good” or better, and almost values were “excellent”, indicating reasonable validity. The POM-Checker showed excellent results with an ICC of 0.9 or higher for all measurements except those for hip FF and extension and left lateral flexion of the lumbar spine, for which the ICC value was between 0.75 and 0.899, indicating good reliability. In the UG measurements, all results for the shoulder and hip joints showed good and excellent reliability, with an ICC value above 0.8. The ICC values for the measurements of FF, Abduction90-ER, and FF90-IR of the shoulder and hip extension were 0.9 or higher, indicating excellent results. The ICC for the FF of the lumbar spine was 0.928, indicating excellent reliability (95% CI = 0.863, 0.963). The other results of the lumbar spine measurements showed fair reliability, with an ICC value between 0.5 and 0.75. The ICC of the POM-Checker measurements was higher than that of the UG measurements for each movement. However, the ICC for the hip FF and lumbar spine FF movements measured using the UG was higher than that of the movements measured by the POM-Checker. Additionally, these insights are similar to those obtained by analyzing the SEM and MDC95 values, indicating absolute reliability.

Regarding inter-test reliability or concordance, the ICC (2,2) value indicated good to excellent reliability of almost all measurements of the movements of the shoulder and lumbar spine. However, for the measurements of FF90-ER of the shoulder, the ICC (2,2) value was 0.708 (95% CI: 0.531, 0.819). Moreover, the hip joint results showed poor reliability, with an ICC (2,2) value of less than 0.5, except for the FF measurements, for which the ICC (2,2) value was 0.73, indicating fair reliability (95% CI: 0.556, 0.269).

## 4. Discussion

Current conventional UG measurements require a skilled medical clinician, and their intra- and inter-test reproducibility remains a problem [37]. To overcome these limitations, a simple and highly accessible measurement device is required. In particular, the device should be compact and provide results that are easy to interpret. Joint ROM is a crucial index in the medical care of the musculoskeletal system. Moreover, most clinicians and patients consider not only the classic anatomical movement concept of maximum significant ROM but also the function (quality) of movement. Therefore, to overcome the limitations of the UG method, we considered RGB-D-based AI measurement methods and evaluated their performance against that of the UG method.

The POM-Checker can objectively assess and perform precise and detailed movement analysis [15]. The POM-Checker does not require operator intervention. This device can measure the maximum ROM value and save images of patients performing movements at the maximum ROM. In addition, it can display the speed of progression of the movement in real time (Figure 6) and is expected to be used for movement function evaluation based on further research.

Reproducibility, which includes reliability and agreement, is a key factor in determining the clinical applicability of measurement devices. Intra-test reliability is important, as the same clinician may take the same measurement on different occasions to document changes. In a clinical setting, more than one tester or measurement device may be involved in ROM measurements, which may lead to further errors. Therefore, intra- and inter-reliability tests should be performed. This study focused on the above-mentioned issues. In measuring the intra-test reliability of the POM-Checker and the UG, the POM-Checker generally showed higher reliability and agreement when evaluating two measurements. The inter- and intra-test reliability was considered poor for ICC values less than 0.50, fair for values between 0.50 and 0.75, good for values between 0.75 and 0.9, and excellent for values greater than 0.90. The ICC for reliability with a value greater than 0.90 ensures reasonable validity. All measurements using both methods were assessed as better than good intra-test reliability when analyzed with ICC. Moreover, the POM-Checker demonstrated excellent reliability in all shoulder measurements. In contrast, most UG measurements showed better reliability than good, but the reliability of the measurements of the lumbar spine movements was only fair. However, the agreement between the results of the two measurement methods, that is, the inter-test reliability, was low in the measurements of some movements, especially those of the hip. In the inter-test reliability analysis of the two methods, the POM-Checker and UG, the ICC value indicated excellent reliability of the results of the lumbar spine movements and mostly good reliability of the results of the shoulder movements. Moreover, the ICC was >0.9, indicating excellent reliability of shoulder abduction and Abduction90-ER measurements. For the hip joint measurements, the ICC indicated fair reliability (0.556 for FF). However, the reliability was poor for all other cases, which exhibited an ICC below 0.5. Additionally, the reliability of the shoulder FF90-ER measurements was fair, with an ICC of 0.708. Consequently, as shown in the design of this study, some limitations remain in the protocol used for measuring active ROM in the standing position, especially for the hip joint.

Despite the above limitations, the results of this study show that the ROM measurement ability of the POM-Checker device is relatively satisfactory. The POM-checker was validated in a previous study for measurements of the shoulder movement [15,25]. However, the POM-Checker cannot be considered technically suitable for ROM measurements because the ICC of intra-test reliability showed good reliability only for hip joint FF, lumbar spine extension, and left lateral flexion movements, with values between 0.75 and 0.90. In addition, the intra-test reliability of the UG was mostly good or excellent, with ICC values above 0.75. It showed excellent results, with an ICC of more than 0.9 for measurements of the shoulder FF, Abduction90-ER, FF90-IR, hip extension, and lumbar spine FF movements. However, the ICC for extension and lateral flexion of the lumbar spine joint showed fair results, ranging from 0.50 to 0.75. These limitations are associated with the use of the method. The UG requires two hands to manipulate the instrument, which can be challenging to position accurately. It also requires clear visual estimation for alignment and measurement reading. When the participant was in the position for performing lumbar spine movements, measurements were performed around the point connecting the pelvis and the central point of the body at shoulder height, between the C7 spinous process and the posterosuperior iliac spine. Bias may have occurred because the area connecting the pelvis and the center of the shoulder was not measured consistently each time. This could be related to the high inter-test reliability of the lumbar spine joint. However, the ICC value for the intra-test reliability of the UG was mostly at a moderate level. Because the motion of the lumbar spine joint is relatively large, it is difficult to determine the exact location of the pivot. Thus, the intra-tester reliability was low for both methods. However, as the participant can perform the spinal joint movements relatively stably, the inter-test reliability was high when comparing the average values of the two methods. The ICC values of inter-test reliability in the lumbar spine joint were greater than 0.9, indicating reasonable reliability.

In inter-test reliability analysis, both the POM-Checker and UG measurement methods showed excellent ICC results for the shoulder and lumbar spine measurements but poor results for the hip joint measurements. This is because the movements of the shoulder and lumbar spine joints can be performed in a relatively stable posture while standing. Thus, the average value of the two measurements is expected to be reliable. The reproducibility of measurements on both methods was high, despite participant movement during the measurements or bias in the POM-Checker and UG instruments. The reliability was low in measurements in which the participants could not stably perform the movements. Furthermore, we noted that inter-test reliability was significantly lower for the hip joint measurements, as the active joint movements were measured in the standing position. When performing hip joint movements while standing on one foot, participants often lose their balance or their bodies do not remain upright, and they tend to lean to one side. For example, when using the POM-Checker for measuring angles, the maximum value of the instantaneous ROM may be recorded without any corrections or considerations. In contrast, when the UG is used, the clinician ensures that measurements are taken only when the participant’s balance is maintained and movement is stopped. The intra-test reliability was high for each method for the hip movement measurements, but the inter-test reliability results showed low reproducibility. Moreover, differences were also significant (*p* < 0.05) in hip joint abduction, adduction, and extension measurements when comparing the mean values of the results of the two measurement methods. The reproducibility of the POM-Checker results was significantly greater for abduction and extension ROM measurements, whereas the reproducibility of the UG results was significantly greater for the hip joint adduction ROM measurements. The SD was relatively large, even though the hip joint possessed a relatively smaller ROM than the other two joints, which might be due to body tilting. The POM-Checker failed to consider the error caused by the body tilt because the results were measured using the angle with respect to the ground and did not measure the actual joint movement range. In other words, when using the UG, the tester took measurements while the participants were balanced, thereby reducing some of the above errors. However, the SD was also large, suggesting that other errors might have occurred when measured using the UG. First, measuring hip joints while standing increases the complexity of maintaining a balanced posture for a period. Thus, even when a clinician calibrates the UG, a tilted posture can result in errors. Second, placing the UG arm in the correct position is difficult, even if it has been adjusted to the reference point for the UG measurement and the vector of the body and legs.

The SEM was less than 1° for all measurements in the intra-test comparison between the POM-Checker and UG results. However, in the inter-test comparison, the SEMs of the ROMs of some shoulder movements and every hip movement were >1°. A comparison of the two groups indicates that, to obtain accurate results, the sample size should be increased. In addition, more detailed measurement standards and protocols are required to reduce the bias, which may be present in this method.

The protocols for measuring the active ROM in the standing position in this study had limitations. Some participants struggled to maintain a stable motion and a constant angle during active ROM measurements of the hip joint in the standing position. Measurement of the active ROM is essential to diagnose problems of the hip joint. However, few studies have been conducted due to the aforementioned challenges. Accurately measuring the active ROM of the hip will require an additional device to stabilize the pelvis while assessing only hip movement. Thus, accurately setting up the apparatus for this method and taking measurements is challenging. The type of movement measured may also affect reliability, and the reliability of active movements may differ from that of passive movements [37].

Furthermore, the 3D RGB-D-based AI ROM measurement method must be improved. Although the orientation of the participant is not relevant in the UG measurement method, the ROM result may be affected when the participant is standing facing the front when using the POM-Checker. For example, when a participant performing a shoulder FF motion is observed from the front, the participant’s elbow or shoulder may be hidden. When the relevant body part is hidden, the algorithm must make a “guess” based on learned data. Thus, the accuracy of the algorithm-based measurements may be lower than that of those obtained through direct observations. For some cases, measuring from the lateral side may be more suitable than measuring from the front; specifically, allowing the camera to capture the movement of the arm without any part of the arm being hidden, can help make the measurement more accurate. The POM-Checker has high-accuracy measurement settings for each movement. However, in this study, to provide consistency during measurement, the measurement process was designed based on the front-facing posture. In addition, this study was conducted in a relatively large indoor location with a white wall in the background. However, the accuracy may decrease to some extent if the measurements are affected by the environment, e.g., unfavorable light conditions or complex background. Several methods have been designed to overcome these limitations, such as installing additional RGB-D cameras in different directions or using larger datasets for AI training. Previously, Mangal et al. [27] reported that Kinect-based ROM measurement requires technical improvement when measuring ROM of the lower extremities compared to when measuring ROM of the upper extremities and proposed a filter to overcome these technical differences.

Several adjustments were made prior to this study. The values measured by the clinician using the UG did not exceed the upper limit of the normal range. This shows a tendency of clinicians to limit the maximum value to the normal maximum range because they are aware of the normal maximum value for joint movement. Some participants can perform FF and abduction movements of the shoulder to an angle greater than 180° without tilting or twisting the body. However, the clinician performing measurements using the UG can be biased due to prior knowledge; for example, 180° is the maximum value of the normal range of shoulder movement. In contrast, the POM-Checker measurements are not affected by previous knowledge. Therefore, prior to this study, we adjusted the AI system of the POM-Checker to consider these aspects. Furthermore, when using the UG, the measurer tends to capture most values in increments of 5°, which are more visible. This might be because the angles on the UG are marked in such increments.

Currently, significant technological advancements have been made to achieve ROM measurements using a 3D RGB-D sensor. However, to apply this sensor to measure ROM in actual clinical practice, protocols similar to those established for using the UG are required. For example, wearing clothes that are not loose, measuring only in positions in which the angle can be stably maintained during ROM, and standardizing landmarks that can be used as reference points on the body surface for each ROM movement. Through continuous comparison between the UG and RGB-D sensor-based methods, a standard protocol appropriate for each measurement method must be established and consent must be obtained. If AI and RGB sensors are technically improved and become more cost-effective, measurement approaches integrating these technologies are expected to be widely implemented for more accurate ROM measurements. Currently, more diverse pose estimator systems are being developed and could be introduced into clinical practice with the support of AI algorithms, which, in turn, would help this technology develop further. In addition, similar methods can be developed to measure the movements of other joints or for clinical applications in other fields, such as for monitoring sarcopenia, spine scoliosis, or the shapes of body parts. For example, it is plausible that in the future, healthcare providers will be able to remotely monitor a patient’s daily ROM progress, potentially helping improve outcomes, while allowing for further development of this technology.

This study has some limitations. First, only a small sample was considered. The use of a larger sample may yield improved results. Second, this study inherently contained potential biases or confounding factors. Most measurements using the UG are usually performed in the supine position [4,19], and other studies have been performed based on these protocols. However, because herein the active ROM was measured in the standing position, the presented results differ from those of previously used methods. Thus, a standard protocol was not followed for the measurement method. This study required participants to repeat a total of eight, four, and four movements of the same shoulder, hip, and lumbar spine, respectively, four times each. However, this had an inevitable bias in that the participant could not create the same angle when repeating the same movement. This bias might have occurred because the participant was asked to repeat the four movements rather than simultaneously take measurements with both the POM-checker and UG. In particular, the repeatability of measurements was low when hip movements were performed in the standing position. This bias may be reduced by allowing a longer rest time for the participants between each measurement or processing the measurement slowly. Alternatively, to improve the analysis, a preliminary study can be conducted in which joint ROM measurements using both methods are performed on a mannequin by setting a fixed angle for the movements. Then, the results of the two methods should be compared. Third, the study was performed on individuals who had normal muscle strength and could balance for a long period when performing joint movements while standing. However, the measurement method has not been tested on patients who are elderly or who cannot perform these movements adequately due to weak muscles or joint disease. Therefore, further research should be conducted on patients with diseases or those who have undergone surgeries related to these ROM analyses. Moreover, studies on whether such participants should have an assistive device to help them balance while standing or whether the ROM should be measured while lying down should be performed.

## 5. Conclusions

This study found that the RGB-D sensor-based AI method can be expected to assist in or complete a UG assessment, which is considered a standard method. The results of this study can be summarized as follows: First, the intra-test reliability of the RGB-D sensor-based AI method was not inferior to that of the UG. Second, compared to UG, the AI-based method shows higher objectivity, and additionally, it has the advantage of saving the ROM process as an image, which can be used later during the follow-up of clinical results. The maximum ROM value and angular velocity can be recorded, eliminating the errors caused by subjective intervention as in the UG method, in which measurements are currently performed by a clinician. Thus, this method can reduce the workforce and time requirements. However, the results of our study did not support the hypothesis that the results of the 3D RGB-D sensor-based AI measurement device are superior to those obtained using the UG. Moreover, this method requires further improvement before it can replace the UG in clinical practice. Limitations of the measurement method protocol or device setting errors can easily lead to biased measurements. Various AI-based methods for measuring ROM using an RGB-D sensor have been introduced. The results of this study suggest that a standardized method should be established in which the posture and movements of the participant being measured are standardized for each joint and validated in accordance with the image recognition and AI analysis methods of the measuring device. These trials are expected to lead to more advanced and systematic ROM measurements.

## Figures and Tables

**Figure 1 medicina-61-00119-f001:**
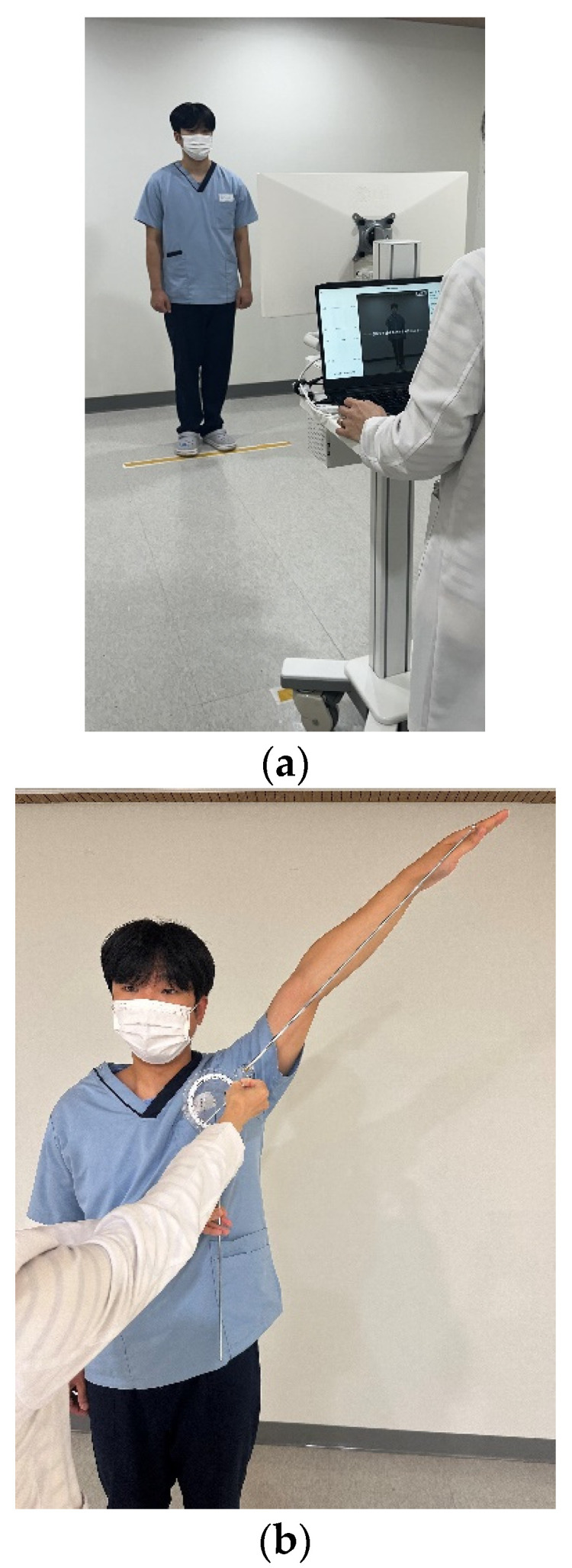
Images of the measurement devices. (**a**) POM-Checker and (**b**) Universal goniometer.

**Figure 2 medicina-61-00119-f002:**
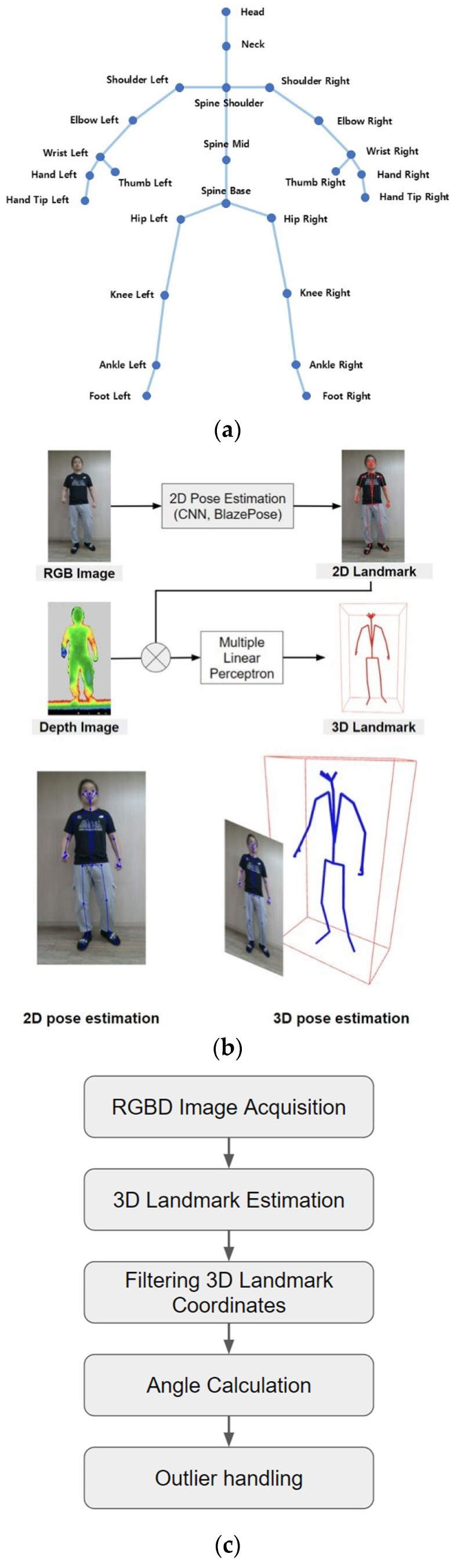
Detailed scheme of the pose estimation algorithm: (**a**) The 25 body joints POM-Checker can capture; (**b**) process of recognizing body joints and estimating the 3D landmarks; (**c**) process of calculating angles.

**Figure 3 medicina-61-00119-f003:**
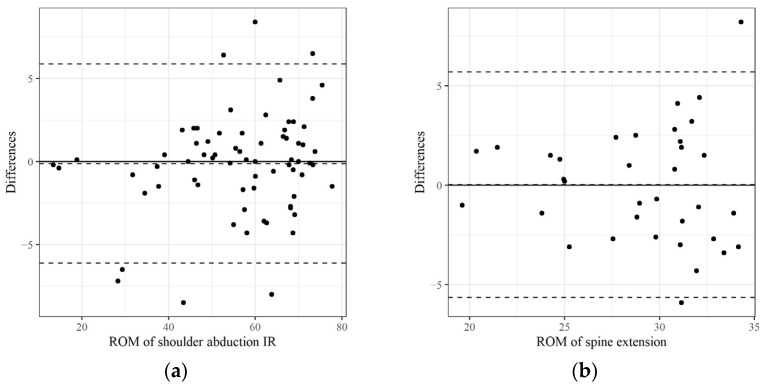
The mean value and difference in the ROM measured twice at each joint using a POM-Checker were graphically represented as a Bland–Altmann plot (intra-test reliabilities): (**a**) The ROM of shoulder abduction IR with the highest ICC; (**b**) the ROM of spine extension with the lowest ICC were selected and compared. IR: Internal rotation; ICC: intraclass correlation coefficient.

**Figure 4 medicina-61-00119-f004:**
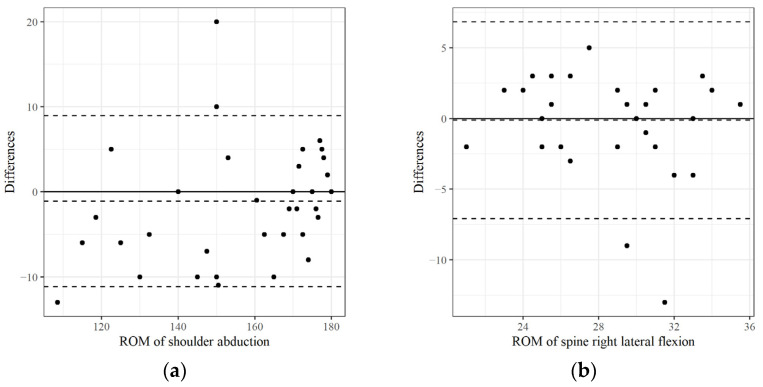
The mean value and difference in the ROM measured twice at each joint using UG were graphically represented as a Bland–Altmann plot (intra-test reliabilities): (**a**) The ROM of shoulder abduction with the highest ICC; (**b**) the ROM of spine right lateral flexion with the lowest ICC were selected and compared. ICC: Intraclass correlation coefficient.

**Figure 5 medicina-61-00119-f005:**
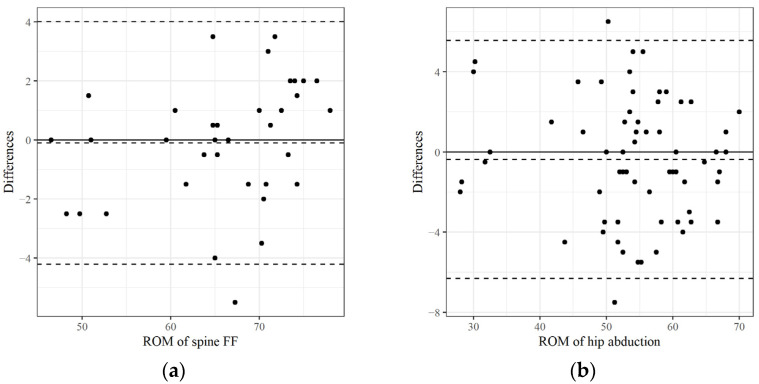
A comparison of the average values measured twice using each device was graphically represented as a Bland–Altmann plot (inter-test reliability): (**a**) The ROM of spine forward flexion with the highest ICC; (**b**) the ROM of hip abduction with the lowest ICC was selected and compared.

**Figure 6 medicina-61-00119-f006:**
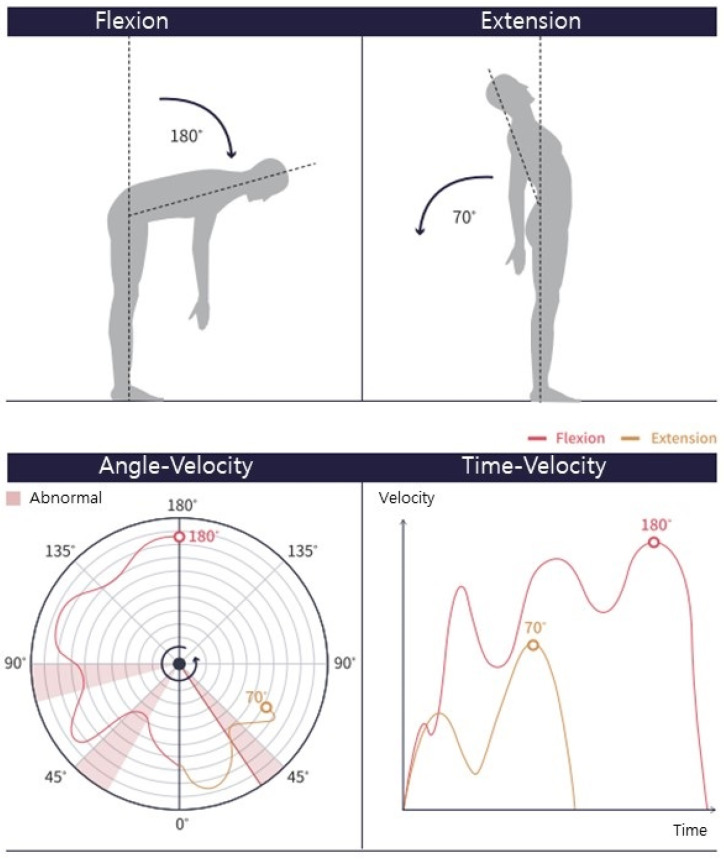
Results of measuring spinal angles using the POM-Checker: the diagram on the left shows the values measured when flexing and extending the spinal joints and the diagram on the right shows the angular velocity of the joint movement over time during the angle measurement process.

**Table 1 medicina-61-00119-t001:** Participant demographic characteristics.

Characteristics (*n* = 35)		Mean	Standard Deviation
Age (year)		38.68	7.89
Body mass index (kg/m^2^)		23.39	2.41
		frequency	%
Sex	Male	15	42.9%
	Female	20	57.1%
Sides evaluated (dominant upper limb)	Right	32	91.4%
	Left	3	8.6%

**Table 2 medicina-61-00119-t002:** POM-Checker intra-test differences and reliability evaluation: Mean value and SD between two measurements.

POM-Checker		1st Measurement		2nd Measurement		*p*-Value (Diff)	Agreement	ICC	95% Confidence Interval (ICC)	SEM	MDC95	95% LOA	Lin (CCC)
		Mean	SD	Mean	SD								
Shoulder (*n* = 70)	FF	163.413	11.232	164.161	10.46	0.684	88.571	**0.941**	(0.907, 0.963)	0.444	1.231	(−8.129, 6.632)	0.939
	Extension	48.189	11.091	48.439	11.455	0.896	91.429	**0.921**	(0.876, 0.95)	0.536	1.486	(−8.992, 8.492)	0.921
	Abduction	159.741	16.877	160.567	17.017	0.774	94.286	**0.945**	(0.913, 0.966)	0.67	1.858	(−11.857, 10.206)	0.944
	Adduction	43.08	9.46	43.37	9.676	0.858	95.714	**0.928**	(0.886, 0.954)	0.435	1.207	(−7.402, 6.822)	0.927
	Abduction ER	74.113	16.654	73.399	17.387	0.804	94.286	**0.952**	(0.923, 0.97)	0.633	1.754	(−9.683, 11.111)	0.951
	Abduction IR	56.4	15.267	56.524	14.619	0.961	90	**0.979**	(0.967, 0.987)	0.366	1.013	(−6.08, 5.832)	0.979
	FF 90 ER	50.904	7.027	51.497	7.281	0.625	95.714	**0.937**	(0.901, 0.961)	0.303	0.84	(−5.659, 4.474)	0.934
	FF 90 IR	68.623	15.633	68.397	16.437	0.934	94.286	**0.967**	(0.948, 0.979)	0.491	1.361	(−7.782, 8.233)	0.967
Hip (*n* = 70)	FF with knee extension	74.341	8.538	74.052	8.869	0.951	95.312	0.874	(0.801, 0.922)	0.546	1.512	(−8.218, 8.796)	0.874
	Extension	43.542	7.111	43.562	7.592	0.988	96.875	**0.911**	(0.857, 0.945)	0.389	1.077	(−6.066, 6.025)	0.911
	Abduction	60.95	9.247	59.675	9.341	0.439	96.875	**0.9**	(0.841, 0.938)	0.518	1.437	(−7.168, 9.718)	0.892
	Adduction	25.528	7.705	25.908	8.205	0.788	92.188	**0.904**	(0.847, 0.941)	0.435	1.206	(−7.191, 6.432)	0.903
Spine (*n* = 35)	FF	78.28	10.405	78.954	9.468	0.778	97.143	**0.923**	(0.853, 0.96)	0.66	1.829	(−8.332, 6.983)	0.921
	Extension	29.051	4.171	29.017	4.251	0.973	94.286	0.765	(0.582, 0.874)	0.489	1.354	(−5.549, 5.618)	0.764
	Left lateral flexion	28.023	3.759	28.84	3.765	0.367	94.286	0.859	(0.739, 0.926)	0.337	0.935	(−4.993, 3.359)	0.839
	Right lateral flexion	28.34	3.751	28.88	3.867	0.555	97.143	**0.905**	(0.821, 0.951)	0.28	0.776	(−3.911, 2.831)	0.896

* Significant difference (*p* < 0.05). The values of ICC that ensure reasonable validity (0.90 or higher) are written in bold letters. SD: Standard deviation; ICC: intraclass correlation coefficient; SEM: standard error of the mean; MDC: minimal detectable change; LOA: limits of agreement; CCC: concordance correlation coefficient; ER: external rotation; IR: internal rotation; FF: forward flexion.

**Table 3 medicina-61-00119-t003:** UG intra-test differences and reliability evaluation; mean value and SD between two measurements.

UG		1st Measurement		2nd Measurement		*p*-Value (Diff)	Agreement	ICC	95% Confidence Interval (ICC)	SEM	MDC95	95% LOA	Lin (CCC)
		Mean	SD	Mean	SD								
Shoulder (*n* = 70)	FF	166.7	11.644	168.086	10.845	0.467	97.143	**0.909**	(0.857, 0.942)	0.574	1.591	(−11.115, 8.344)	0.902
	Extension	49.671	9.297	50.871	8.003	0.415	94.286	0.835	(0.747, 0.894)	0.595	1.65	(−11.172, 8.772)	0.827
	Abduction	165.614	19.77	166.7	17.936	0.734	95.714	**0.963**	(0.941, 0.977)	0.614	1.701	(−11.302, 9.131)	0.961
	Adduction	41.443	10.562	43.4	11.445	0.295	98.571	0.891	(0.83, 0.931)	0.615	1.705	(−12.68, 8.765)	0.877
	Abd ER	73.743	16.936	73.157	17.014	0.839	97.143	**0.949**	(0.919, 0.968)	0.648	1.796	(−10.024, 11.195)	0.948
	Abd IR	60.4	12.442	61.086	12.599	0.746	98.571	0.872	(0.802, 0.918)	0.757	2.099	(−13.086, 11.715)	0.871
	FF90 ER	52.829	6.884	52.986	7.062	0.894	98.571	0.836	(0.749, 0.895)	0.477	1.321	(−7.923, 7.609)	0.836
	FF90 IR	72.243	13.926	71.943	14.071	0.899	94.286	**0.911**	(0.861, 0.944)	0.705	1.954	(−11.193, 11.793)	0.911
Hip (*n* = 70)	FF with knee extension	76.625	8.851	76.891	8.471	0.863	98.438	0.879	(0.809, 0.925)	0.532	1.474	(−8.556, 8.025)	0.879
	Extension	34.547	8.236	33.797	9.47	0.633	96.875	**0.9**	(0.84, 0.938)	0.497	1.376	(−7.113, 8.613)	0.897
	Abduction	57.312	6.949	57.641	6.627	0.785	100	0.802	(0.694, 0.875)	0.534	1.479	(−8.655, 7.998)	0.801
	Adduction	30.938	8.02	31.219	7.38	0.837	98.438	0.881	(0.811, 0.926)	0.471	1.304	(−7.623, 7.061)	0.88
Spine (*n* = 35)	FF	78.6	9.153	79.029	9.272	0.846	97.143	**0.928**	(0.863, 0.963)	0.59	1.636	(−7.226, 6.369)	0.927
	Extension	29.029	4.592	29.229	4.659	0.857	100	0.73	(0.527, 0.854)	0.575	1.594	(−6.784, 6.384)	0.729
	Left lateral flexion	29	3.726	29.2	3.53	0.818	100	0.71	(0.497, 0.842)	0.467	1.295	(−5.552, 5.152)	0.709
	Right lateral flexion	28.6	3.767	28.714	4.163	0.904	94.286	0.601	(0.339, 0.777)	0.599	1.661	(−6.968, 6.74)	0.601

* Significant difference (*p* < 0.05). The values of ICC that ensure reasonable validity (0.90 or higher) are written in bold letters. SD: Standard deviation; ICC: intraclass correlation coefficient; SEM: standard error of the mean; MDC: minimal detectable change; LOA: limits of agreement; CCC: concordance correlation coefficient; ER: external rotation; IR: internal rotation; FF: forward flexion.

**Table 4 medicina-61-00119-t004:** Comparison of the average values measured twice using each device; inter-test differences and reliability evaluation.

Mean Value		POM-Checker		UG		*p*-Value (Diff)	Agreement	ICC	95% Confidence Interval (ICC)	SEM	MDC95	95% LOA	Lin (CCC)
		Mean	SD	Mean	SD								
Shoulder (*n* = 70)	FF	163.787	10.692	167.393	10.992	0.051	95.714	0.874	(0.797, 0.922)	0.867	2.404	(−19.396, 12.185)	0.735
	Extension	48.314	11.049	50.271	8.309	0.238	92.857	0.765	(0.621, 0.854)	1.02	2.826	(−18.997, 15.082)	0.607
	Abduction	160.154	16.714	166.157	18.7	0.047	92.857	**0.927**	(0.883, 0.955)	1.102	3.055	(−27.462, 15.457)	0.817
	Adduction	43.225	9.394	42.421	10.708	0.638	91.429	0.787	(0.657, 0.867)	1.01	2.798	(−15.709, 17.316)	0.646
	Abd ER	73.756	16.818	73.45	16.757	0.914	91.429	**0.941**	(0.904, 0.963)	0.95	2.633	(−15.171, 15.782)	0.888
	Abd IR	56.462	14.868	60.743	12.114	0.064	95.714	0.815	(0.702, 0.885)	1.282	3.552	(−26.77, 18.208)	0.654
	FF90 ER	51.201	7.042	52.907	6.682	0.144	92.857	0.708	(0.531, 0.819)	0.78	2.161	(−14.834, 11.421)	0.532
	FF90 IR	68.51	15.908	72.093	13.684	0.155	94.286	0.858	(0.771, 0.912)	1.251	3.469	(−25.134, 17.969)	0.729
Hip (*n* = 70)	FF with knee extension	74.196	8.427	76.758	8.397	0.087	96.875	0.556	(0.269, 0.73)	1.166	3.233	(−21.389, 16.265)	0.368
	Extension	43.552	7.189	34.172	8.65	0	96.875	0.268	(−0.206, 0.555)	1.293	3.583	(−17.869, 36.63)	0.091
	Abduction	60.312	9.06	57.477	6.445	0.044	92.188	0.048	(−0.567, 0.422)	1.373	3.804	(−19.228, 24.9)	0.023
	Adduction	25.718	7.766	31.078	7.473	0	95.312	0.253	(−0.229, 0.546)	1.246	3.453	(−27.404, 16.684)	0.116
Spine (*n* = 35)	FF	78.617	9.754	78.814	9.046	0.93	97.143	**0.975**	(0.95, 0.987)	0.499	1.384	(−5.919, 5.524)	0.95
	Extension	29.034	3.956	29.129	4.302	0.924	91.429	**0.92**	(0.842, 0.96)	0.379	1.051	(−4.432, 4.243)	0.852
	Left lateral flexion	28.431	3.627	29.1	3.356	0.426	91.429	**0.908**	(0.817, 0.953)	0.343	0.951	(−4.803, 3.466)	0.816
	Right lateral flexion	28.61	3.718	28.657	3.552	0.957	94.286	**0.915**	(0.832, 0.957)	0.344	0.953	(−3.976, 3.882)	0.844

* Significant difference (*p* < 0.05). The values of ICC that ensure reasonable validity (0.90 or higher) are written in bold letters. SD: Standard deviation; ICC: intraclass correlation coefficient; SEM: standard error of the mean; MDC: minimal detectable change; LOA: limits of agreement; CCC: concordance correlation coefficient; ER: external rotation; IR: internal rotation; FF: forward flexion.

## Data Availability

The data presented in this study are available on request from the corresponding author. The data are not publicly available due to privacy or ethical restrictions.

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
