# Peer review of "Exploring the Potential of AI-Assisted Technology in Joint Range-of-Motion Measurements: A Reliability Study"

_medicina, 2025, doi:10.3390/medicina61010119_

Round 1
Reviewer 1 Report
Comments and Suggestions for Authors
This is an interesting study examining the utility of a 3D RGB-D sensor AI device, POM Checker (Elysium Inc., Seoul, Republic of Korea). The measurements obtained from the POM-Checker have been compared to those obtained using a goniometer, resembling a validation study for the computer vision-based system. The study concludes that the results did not show that the measurements from the POM-Checker are superior to those obtained using the goniometer. However, authors point out certain additional benefits to using the computer vision-based system, as it offers greater objectivity and the advantage of saving the ROM process as an image or video for future reassessment.
I have a few questions:
The authors mention that the camera was always positioned in the front. However, for measuring lumbar spinal movement, it seems that the camera needs to be positioned at the side, as shown in Figure 3. Could you please clarify this?
Is POM an abbreviation for something? I couldn't find this information in the manuscript.
Can a few Bland-Altman plots be shown to visualize the agreement or consistency between two measurement methods? For example, the authors could show one plot for the measurement with the highest ICC and another for the measurement with the lowest ICC. This would result in 2 plots for each of Tables 2, 3, and 4, for a total of 6 plots.
I believe a Future Directions section is needed, as there are multiple use cases for training the AI model to facilitate different tasks, both diagnostic and therapeutic. For example, a video could be displayed on a mobile device showing someone performing a joint range of motion (ROM) exercise, which the patient would then repeat in front of a camera so that measurements can be taken. This would become a tool that can be used at home without the need for assistance in taking measurements. The healthcare provider could remotely monitor the patient's daily ROM progress. I believe this is the direction the field is moving on, with single-camera systems and AI algorithms paving the way for this paradigm shift. This could be discussed in the article.
Various camera systems are currently available, including both single-camera and multi-camera systems, as well as open-source algorithms such as MediaPipe and TensorFlow Pose Detection from Google, MMPose from MMLab, OpenPose from CMU, and AlphaPose from SJTU. These algorithms are used for converting video to skeleton data. These can be mentioned in the Future Directions section, as many systems are being developed using them. Of course, if the authors are aware of further developments in the field, they can be included to make the discussion more comprehensive and forward-looking.
Author Response
Comments 1:
This is an interesting study examining the utility of a 3D RGB-D sensor AI device, POM Checker (Elysium Inc., Seoul, Republic of Korea). The measurements obtained from the POM-Checker have been compared to those obtained using a goniometer, resembling a validation study for the computer vision-based system. The study concludes that the results did not show that the measurements from the POM-Checker are superior to those obtained using the goniometer. However, authors point out certain additional benefits to using the computer vision-based system, as it offers greater objectivity and the advantage of saving the ROM process as an image or video for future reassessment.
I have a few questions:
The authors mention that the camera was always positioned in the front. However, for measuring lumbar spinal movement, it seems that the camera needs to be positioned at the side, as shown in Figure 3. Could you please clarify this?
Is POM an abbreviation for something? I couldn't find this information in the manuscript.
Can a few Bland-Altman plots be shown to visualize the agreement or consistency between two measurement methods? For example, the authors could show one plot for the measurement with the highest ICC and another for the measurement with the lowest ICC. This would result in 2 plots for each of Tables 2, 3, and 4, for a total of 6 plots.
I believe a Future Directions section is needed, as there are multiple use cases for training the AI model to facilitate different tasks, both diagnostic and therapeutic. For example, a video could be displayed on a mobile device showing someone performing a joint range of motion (ROM) exercise, which the patient would then repeat in front of a camera so that measurements can be taken. This would become a tool that can be used at home without the need for assistance in taking measurements. The healthcare provider could remotely monitor the patient's daily ROM progress. I believe this is the direction the field is moving on, with single-camera systems and AI algorithms paving the way for this paradigm shift. This could be discussed in the article.
Various camera systems are currently available, including both single-camera and multi-camera systems, as well as open-source algorithms such as MediaPipe and TensorFlow Pose Detection from Google, MMPose from MMLab, OpenPose from CMU, and AlphaPose from SJTU. These algorithms are used for converting video to skeleton data. These can be mentioned in the Future Directions section, as many systems are being developed using them. Of course, if the authors are aware of further developments in the field, they can be included to make the discussion more comprehensive and forward-looking.
Response:
Comment 1: The authors mention that the camera was always positioned in the front. However, for measuring lumbar spinal movement, it seems that the camera needs to be positioned at the side, as shown in Figure 3. Could you please clarify this?
Response 1: Thank you for your comment and bringing to our attention aspects of our study that should be presented more clearly. To present the angle change, we provided a side view in Figure 6. However, the measurements in this study, as reported in the manuscript, were performed with participants front-facing the camera. These measurements involved observing points on the subject’s upper body, as they moved closer to and farther away from the camera. Side measurements may have improved the accuracy; however, we compensated for any inaccuracies by using an AI algorithm, as described in the manuscript (lines 551 to 566).
Comment 2: Is POM an abbreviation for something? I couldn't find this information in the manuscript.
Response 2: POM is a portmanteau of posture and ROM. This has now been explained in line 232.
Comment 3: Can a few Bland-Altman plots be shown to visualize the agreement or consistency between two measurement methods? For example, the authors could show one plot for the measurement with the highest ICC and another for the measurement with the lowest ICC. This would result in 2 plots for each of Tables 2, 3, and 4, for a total of 6 plots.
Response 3: We added the content you mentioned as figures in the results section. (Figures 3, 4, and 5) This has now been explained in lines 378 to 380.
Comment 4: Future Directions section
Response 4:
1. Currently, more diverse pose estimator systems are being developed and could be introduced into the clinical practice with the support of AI algorithms, resulting in their further refinement (lines 592 to 594).
2. For example, it is plausible that in the future healthcare providers will be able to remotely monitor a patient's daily ROM progress, potentially helping improve outcomes, while allowing for further development of this technology (lines 597 to 599).

Reviewer 2 Report
Comments and Suggestions for Authors
Dear Authors
The paper is well- structured, clearly written and effectively compares the traditonal used methods of measuring joint range of motions to those asissted by AI. The analysis is robust, the conclusion drawn are well- suported by the data presented. additionally, the manuscript adheres to the journals guidelines and meets the necessary standards for quality and originality.
The manuscript addresses the reliability of the newly introduced RGB-D sensor-based artificial intellegence-assisted POM Checker device for measuring the range of motion in the shoulder, hip, and lumbar spine, as well as comparing the results with the corresponding values obtained using traditional methods of the universal goniometer. Previous studies have validated the new technology in the shoulder joint; however, it has not been tested in the hip joint and lumbar spine. The current study filled this gap and addressed inter-adn intra-test reliability in these three locations.
The manuscript is well structured, the introduction is informative, the methods and materials are well designed and described, and the results are detailed. The references are appropriate, and the figures are demonstrative.
This conclusion is supported by the results, and presents valuable insights into the accuracy and reliability of the device. This paper can be regarded as one of the poineer studies that would be followed by further studies that aid in the advancement and refinement of the technology.
Author Response
Thank you for your comments.